# Liver Kinase B1 Functions as a Regulator for Neural Development and a Therapeutic Target for Neural Repair

**DOI:** 10.3390/cells11182861

**Published:** 2022-09-14

**Authors:** En Huang, Shuxin Li

**Affiliations:** Shriners Hospitals Pediatric Research Center, Department of Neural Sciences, Lewis Katz School of Medicine, Temple University, Philadelphia, PA 19140, USA

**Keywords:** liver kinase B1, neuronal polarity, intrinsic growth capacity, CNS injury, axon regeneration, functional recovery

## Abstract

The liver kinase B1 (LKB1), also known as serine/threonine kinase 11 (STK11) and Par-4 in *C. elegans*, has been identified as a master kinase of AMPKs and AMPK-related kinases. LKB1 plays a crucial role in cell growth, metabolism, polarity, and tumor suppression. By interacting with the downstream signals of SAD, NUAK, MARK, and other kinases, LKB1 is critical to regulating neuronal polarization and axon branching during development. It also regulates Schwann cell function and the myelination of peripheral axons. Regulating LKB1 activity has become an attractive strategy for repairing an injured nervous system. LKB1 upregulation enhances the regenerative capacity of adult CNS neurons and the recovery of locomotor function in adult rodents with CNS axon injury. Here, we update the major cellular and molecular mechanisms of LKB1 in regulating neuronal polarization and neural development, and the implications thereof for promoting neural repair, axon regeneration, and functional recovery in adult mammals.

## 1. Introduction

The liver kinase B1 (LKB1), also known as serine/threonine kinase 11 (STK11, Par-4 in *C. elegans*), was first identified in its mutated form with loss of function in Peutz–Jeghers syndrome (PJS), a rare dominantly inherited autosomal disease. Germline inactivating mutations of LKB1 are associated with the pathogenesis of PJS characterized by multiple benign polyps in the gastrointestinal system and numerous metastatic malignancies [1]. Because patients with PJS have a significantly increased risk of developing cancer, LKB1 is recognized as a tumor suppressor. Heterozygous knockout mice with downregulated LKB1 develop tumors, primarily hepatocellular carcinoma, spontaneously and late in life [2]. LKB1 mutations are also linked to extraintestinal cancers, including lung cancer, breast cancer, and cervical carcinomas [3].

LKB1 is essential for controlling the metabolism, growth, and polarity of various types of cells [4], including neuronal polarity, axon formation, and axon extension. In response to stimulation by extracellular factors, including BDNF, NGF, Sema3A, netrin-1, Reelin, and Wnt, LKB1 acts as the downstream effector of cAMP/PKA and PI3 kinase, and is a major determinant for axon differentiation by regulating neuronal migration, axon initiation, and axon elongation in the CNS. LKB1 deletion or downregulation eliminates axon formation in vivo; its overexpression stimulates the formation of multiple axons. As a master upstream kinase of the multiple signals that control cell growth, LKB1 controls axon development largely by activating AMPK-related kinases, especially SAD-A, SAD-B, and NUAK1 kinases.

Given the critical roles of LKB1 in controlling axon genesis and elongation, this kinase could be important to mediating axon regeneration and neural repair in the adult nervous system after injury. Recent studies demonstrate that LKB1 upregulation promotes robust and long-distance regrowth of injured descending motor tracts into the caudal spinal cord in adult rodents with spinal cord injury (SCI) [5]. This review focuses on the cellular and molecular mechanisms of LKB1-mediated neuronal polarization and brain development, and recent progress in targeting LKB1 for stimulating axon regeneration, neural repair, and functional recovery after CNS injury.

## 2. LKB1 Gene, Structure, and Overall Cellular Functions

The human LKB1 gene (23 kb) is composed of 10 exons on chromosome 19p13.3 [6]. LKB1 has various homologs in mouse, Drosophila, Xenopus (called XEEK1), and *C. elegans* (called PAR-4) [7,8,9,10]. Despite the unknown significance, LKB1 mRNA has several splice variants of 1302 bp and 444 bp in-frame deletion of exons 5–7 and part of exon 8, and a variant retaining intron [11,12]. Human LKB1 protein (436 amino acids, MW: ~50 kDa) consists of an N-terminal non-catalytic domain with two nuclear localization signals, a kinase domain at N-terminus (aa 49-309, close to AMPK members), and a C-terminal regulatory domain with conserved phosphorylation sites [4,12]. 

LKB1 is an important serine/threonine kinase required for maintaining cell metabolism, energy homeostasis, and polarity by activating AMPK (Thr-172) and ~12 other AMPK-related kinases. Numerous highly conserved residues on LKB1 are phosphorylated either by auto-phosphorylation (Thr-185, Thr-189, Thr-336, and Ser-404) or by upstream kinases (Ser-31, Ser-325, Thr-366, and Ser-431). Activation of LKB1 requires the formation of a complex in the nucleus with its cofactors STE20-related kinase adaptor alpha (STRADɑ) and MO25 (also known as calcium-binding protein 39, Figure 1). After activation and translocation to the cytoplasm, LKB1 is phosphorylated at Ser431 by active cytoplasmic PKA, PKCζ, and S6 kinases, and stimulates the active transport of LKB1 to the cytoplasm [13]. Among the numerous phosphorylation sites, phosphorylation at Ser431 by PKA or p90RSK is especially important for controlling LKB1 functions, including cell cycle management, polarity formation, and axon specification. LKB1 activates various downstream signaling pathways, including AMPK, SAD (synapses of amphids defective kinase), NUAKs, and other kinases [14]. 

LKB1 is widely expressed in many embryonic and adult tissues, including the developing brain, with the highest level in the forebrain [15,16]. LKB1 is concentrated in the cell nucleus and is also present in the axons of cortical neurons. Its expression pattern is similar to that of its cofactors STARD and MO25. They usually form an LKB1–STARD–MO25 complex to function. Integration of LKB1 with various environmental signals activates downstream signaling and regulates the functions of various types of cells, including controlling glucose/lipid metabolism and cell polarity, and suppressing the growth of cancer cells. LKB1 is critical to the polarization of epithelial cells and axon formation of developing neurons by regulating the distribution of the Golgi apparatus in the cytoplasm. Its function in the adult nervous system is largely unknown although it is required for myelination of peripheral axons by Schwann cells [17,18,19]. LKB1 is also critical to the differentiation of neural crest stem cells [20] and polarization of epithelial cells in mammals [21,22].

## 3. LKB1 Forms Protein Complexes for Function in Neural Cells

LKB1 functions as a critical convergent signal downstream of numerous extracellular factors and transmembrane receptors. The binding of various extracellular molecules to their receptors and the intrinsic asymmetry of cytoplasmic components may activate the cAMP/PKA, Ras/ERK, and Ras/PI3K signaling pathways, alter the activities of downstream signals and regulate diverse cellular functions. Among the diverse signals downstream of these pathways, phosphorylating LKB1 at different sites is crucial to cell functions [23]. Particularly, neurotrophins interact with their receptor tyrosine kinases and stimulate neuronal growth by activating the cAMP/PKA, Ras/ERK, and Ras/PI3K/Akt signals [24,25], all of which could modulate LKB1 phosphorylation and activity (Figure 2) [26,27].

LKB1 monomer is primarily located in the nucleus and is largely inactive, but it usually forms a heterotrimeric complex with STRADα and MO25, which stabilize LKB1 and activate its kinase activity. The function and localization of LKB1 are highly associated with its binding to STRAD, and this binding is essential for phosphorylating AMPK-related enzymes by LKB1 [4]. The binding of LKB1 to STRAD alters LKB1 conformation and enhances its activity 100-fold [28]. The removal of endogenous STRADα by small interfering RNA abolishes the LKB1-induced G1 phase arrest [29]. LKB1 and STRADα have a reciprocal protein-stabilizing relationship in vivo and STRADα maintains LKB1 protein levels specifically by cytoplasmic compartmentalization [30]. The ubiquitously expressed scaffolding MO25 (mouse protein 25) is the third component of the LKB1–STRAD–MO25 complex in a similar stoichiometry, and it interacts with STRADα C-terminal and further stabilizes the binding of LKB1 to STRADα [28,31]. Upon interaction with STRAD and MO25 in the nucleus, LKB1 is translocated to the cytoplasm from the nucleus for functioning [12,29].

Atypical PKC (aPKC) complexes, including PKCζ/Par6/Par3 and PKCζ/Par6/Par3/CDC-42, probably also activate LKB1 in neurons and other types of cells. The aPKC complexes are required for proteoglycan-induced axon growth inhibition [32]. Out of several components in the aPKC complex, PKCζ is a member of the aPKC subfamily, phosphorylates LKB1 at Ser-428/431, and mediates the activation of AMPK in endothelial cells [33,34]. Activating LAR receptors by CSPG application reduces the activities of PKCζ and LKB1, indicating that PKCζ-mediated LKB1 suppression also mediates axon growth inhibition by the scar-scoured inhibitor CSPGs [35]. Other signaling proteins may also interact with LKB1 and modulate its relocation and activity. For example, the interactions between LKB1 and PTEN have been shown to promote LKB1 relocation to the cytoplasm in cancer cells [36].

## 4. Major Signaling Pathways Downstream of LKB1 in Neurons or Other Cell Types

LKB1 has a sequence similarity to the catalytic domain of AMPK and functions as a master kinase of AMPKα/β and 12 AMPK-related kinases, including SIK1-3, MARK1-4, NUAK1/2, SAD-A/B, and SNRK (SNF Related Kinase, Figure 3). The activation loops in AMPK and AMK-related kinases are homologous [12,37] and are phosphorylated and activated by LKB1 [37]. LKB1 activates AMPK by efficiently phosphorylating Thr172 in its activation loop [28,38]. In response to reduced levels of ATP and subsequent increase of AMP, AMPK serves as an energy-sensor to switch off cell metabolism, and prevents cell growth and proliferation in nutrient deficient conditions [39]. AMPK regulates physiological processes by phosphorylating several transcription factors, including CREB and its related family members of HDACs [40]. AMPK controls cell polarity and cytoskeletal dynamics [39,40] also by directly phosphorylating the motor protein KLC2 [41] and microtubule-associated protein Tau [42].

SIK1-3 isoforms are the feedback regulators of cAMP-mediated gene expression. SIK1/2 phosphorylates CRTC2 (CREB regulated transcription coactivator 2) and suppresses CREB [43]. SIK2/3 phosphorylates class IIa HDACs and alters their activity, thus regulating various physiological and pathological cellular programs [44,45]. SIK2/3 also stimulates the binding of IIa HDACs to 14-3-3 protein and the nucleocytoplasmic trafficking of class IIa HDACs. Heterozygous SIK1 mutations significantly downregulate ARC (activity-regulated cytoskeleton-associated protein) and other synaptic activity response element genes and may cause SIK1 syndrome, a developmental epilepsy disorder [46]. Moreover, as the LKB1 substrates [47], MARKs are critical for maintaining the polarity of epithelial and neural cells. MARKs phosphorylate MAPs (e.g., MAP2, MAP4, and Tau), dissociate them from microtubules, and increase the dynamic instability of microtubules [48]. Expression of MARK2 alters neuronal migration and transits multipolar cells to bipolar cells in the developing cerebral cortex, an essential step during neuronal polarization in vivo [49].

SAD kinase A and B (also known as brain-specific kinase 1 or 2, BRSK1/2) are additional AMPK-related kinases largely restricted to the nervous system with broad distribution in the brain and spinal cord of embryonic and postnatal animals [50]. Activated SAD kinases are important for controlling the cell cycle, energy metabolism, centrosome duplication, and neural development by phosphorylating the MAP Tau and regulating microtubule dynamics during neuronal polarization. SAD B also regulates neurotransmitter release in neurons.

NUAKs regulate cell adhesion, ploidy, proliferation, and senescence. NUAKs are critical for reorganizing cytoskeletal structures in neuroepithelial cells during neurulation. NUAK1 phosphorylates myosin phosphatase targeting-1 (MYPT1), which phosphorylates myosin light chain 2 (MLC2) and activates non-muscle myosin II [51,52,53], the essential cytoskeletal proteins in migratory cell polarity [54]. Double mutants of NUAK 1 and 2 caused neural tube defects due to altering the distribution of phosphorylated MLC2, F-actin, and acetylated α-tubulin-positive microtubules [55].

LKB1 directly phosphorylates the Thr(109) of PAK1 (also known as p21-activated kinase 1), which mediates LKB1-induced suppression of cell migration [56]. PAK1 also acts as a substrate of Akt [57] and phosphorylates and regulates several cytoskeleton signaling proteins, including the Arp2/3 complex [58] and LIM-kinase [59]. In addition, LKB1 activates SNRK by phosphorylating its Thr173 in the T-loop in the presence of STRAD and MO25 [60]. SNRK potentially mediates hematopoietic cell proliferation or differentiation and neuronal apoptosis.

## 5. Critical Role of LKB1 in Neuronal Polarity and Axon Development

LKB1 functions as a major determinant for neuronal polarization by regulating neuronal migration, axon-forming neurite initiation, and axon extension [14,61]. Neurons are highly polarized, and establishing neuronal polarity and asymmetrical distribution of cellular components within a neuron (i.e., the formation of a single axon and multiple dendrites) is essential for nervous system development [62]. The high metabolic demands and integrity of extremely long structures of axons require chemical and signaling support from neuronal somas and surrounding glia, including Schwann cells (SCs) in PNS and oligodendrocytes in CNS [63]. Our understanding of molecular events underlying neuronal polarization is largely based on studies with dissociated embryonic neuronal cultures, especially the hippocampal neurons [64], which are well characterized by 4 stages during polarization (1–4).

PKA-dependent LKB1 phosphorylation represents an early signal for axon formation and is essential for neuronal polarization [14]. During development, cAMP/PKA and PI3K pathways are critical for controlling neuronal growth in response to stimulation by various extracellular factors (BDNF, NGF, Sema3A, netrin-1, Reelin, and Wnt); these pathways are activated in parallel and converge on common downstream effectors that regulate axon differentiation. Among various extracellular molecules, the repulsive Sema3a suppresses axon formation in developing neurons. Locally applying Sema3A promoted dendrite formation of cultured hippocampal neurons, but suppressed the differentiation of undifferentiated neurite into axon [65]. Bath application of Sema3A to polarized hippocampal neurons promoted dendrite growth but suppressed axon elongation. Sema3A antagonizes axon genesis and extension by reducing PKA-dependent phosphorylation of LKB1 and GSK-3β [65].

LKB1 deletion or downregulation eliminates axon formation during development in vivo. Mice lacking LKB1 died between E9 and E10 before the cerebral cortex formation [66]. Selective deletion of LKB1 in dorsal telencephalic progenitors (the source for cortical pyramidal neurons) in the Emx1-Cre line did not affect the migration of pyramidal neuron progenitors in the cerebral cortex, but caused a striking absence of corticofugal and callosal axons in mice [15,61]. LKB1 knockdown by RNAi electroporation in utero in E18 rats prevented axon specification [61] and neuronal migration in vivo [67]. Overall, LKB1 mainly has an early role in establishing polarity because deleting LKB1 in postmitotic neurons of Nex-Cre mice did not cause obvious polarity defects [68].

Accumulation of LKB1 and STRAD correlates with axon differentiation and their overexpression stimulates multiple axon formation. Various extracellular polarizing signals and/or intrinsic factors activate signaling pathways, including local elevation of cAMP levels, and phosphorylate LKB1, which forms the complex with STRAD and MO25 to stabilize LKB1 and its kinase activity. The mRNAs for LKB1, STRAD, and MO25 are broadly expressed E15.5 in the developing brain, with the highest levels in neuronal progenitors and postmitotic neurons of the forebrain [15]. Upregulation of these three complex genes is maintained throughout embryogenesis and postnatal life [15,61]. Phosphorylated LKB1 at S431 is concentrated in the axon of E15.5 neurons while total LKB1 is present at similar levels in all neurites at this stage. In mature hippocampal neuronal cultures, LKB1 and STRAD are mostly localized to differentiated axons [15,61].

The PAR3/PAR6/aPKC or PAR3/PAR6/PKCζ complex has a conserved role in establishing polarity in certain types of eukaryotic cells, including some neurons [32]. LKB1 is phosphorylated by αPKC and PKCζ at Ser 428 in endothelial cells [33,69]. PI3-kinase activation localizes the PAR3/PAR6/aPKC complex with CDC42 at the tips of growing axons and regulates neuronal polarization [70,71,72]. The binding of Cdc42-GTP to PAR6 determines the localization of the PAR3/PAR6/aPKC complex and the subsequent activation of Rac1 [73]. Rac1 also activates PI3-kinase [74] and forms a positive feedback loop. As two small GTPases in the Rho family, CDC42 and Rac1 activate multiple specific downstream effectors for cytoskeletal reorganization and axon specification [75].

LKB1 influences neuronal polarity through several downstream kinases linked to cytoskeletal remodeling, including SAD, NUAK, and MARKs [15,61,68]. LKB1 determines axon development largely by activating the PAR-1-related kinases, especially SAD-A and SAD-B, which are essential for controlling axon differentiation. Local phosphorylation of LKB1 on S431 phosphorylates and activates SAD-A and SAD-B at T-loop residue Thr-175 and Thr-187, respectively [15]. SAD-A and -B are principally expressed in the brain and spinal cord of embryonic and postnatal mice [50]. Phosphorylated SAD is concentrated in the cortical axons at E15.5 [15]. SAD-dependent neuronal polarity in vivo is cell type-specific [76] because SAD kinases are required at the early stages of axon formation in cortical pyramidal neurons, but not in the spinal cord or brain stem. SAD kinases also contribute to axon branching in some sensory neurons at a later stage. SAD-A and -B can phosphorylate the axon protein Tau at S262 [77], which controls Tau binding to microtubules and subsequent microtubule organization [15,50]. Double SAD-A and -B knockout mice exhibit abnormally oriented dendrites and a striking deficiency of cortical axon formation in vivo [50]. Tsc1 and Tsc2, the signals downstream of PI3K, can also regulate SAD activities by locally restricting their translation [78]. SAD kinases thus represent a convergent point where PKA/LKB1 and PI3K/Akt pathways regulate axon formation by phosphorylating Tau.

Among the other substrates of LKB1, NUAK1 can control axonal mitochondria immobilization and branching [68]. Postmitotic conditional deletion of LKB1 or knockdown of NUAK1 after axon specification drastically reduces axon branching in vivo, whereas their overexpression sufficiently augments axon branching. LKB1 deletion reduced NUAK1 level and activity in cultured neurons. The LKB1–NUAK1 pathway regulates axonal terminal branching and arborization by immobilizing mitochondria, specifically at nascent presynaptic sites. Therefore, LKB1 controls at least two kinase pathways in cortical neurons during development with high temporal and spatial specificity, SAD-A/B for axon specification at an early stage of axon development, and NUAK1 for axon growth and branching at postnatal stages. It will be interesting to further dissect how LKB1 regulates both SAD-A/B and NUAK1 pathways during neurodevelopment.

MARK2 is also a substrate of LKB1 and has been implicated in neuronal polarity [79,80]. Appropriately regulating MARK2 expression levels is essential for neuronal polarization in vivo [75]. MARK2 negatively regulates axon formation downstream of the PAR-3/PAR-6/aPKC complex [80] because its knockdown promotes axon formation and outgrowth. MARK2 phosphorylates MAPs (e.g., tau, MAP2/4, Tau1, and doublecortin) and reduces their stability by dissociating them from microtubules. Deleting MARK2 did not display obvious neural developmental defects in mice [81], probably because of the compensation of other MARKs following MARK2 deletion. It will be interesting to link MARK2-mediated axon development to LKB1 activation.

## 6. LKB1 Regulates Schwann Cell (SC) Function and PNS Myelination during Development

LKB1 is crucial to regulating the polarity of PNS ensheathing SCs and PNS myelination. An in vitro study using SC–DRG co-cultures indicated that cAMP-PKA-dependent phosphorylation of LKB1 in SCs is required to establish asymmetric localization of LKB1 and PAR-3 on the SC-axon interface [18]. In SC–DRG co-cultures, selective LKB1 deletion in SCs delayed SC maturation, myelination initiation, and myelin thickness, while expressing WT or S431D phosphomimetic LKB1, but not S431A, in LKB1-deleted SCs, rescued the asymmetric localization of LKB1 and PAR-3, and the myelination phenotype [18]. SC-specific deletion of LKB1 in CNP-Cre cKO mice disrupted asymmetric localization of PAR-3 and significantly attenuated developmental myelination without altering SC proliferation or alignment in vivo. Moreover, LKB1 is required for maintaining stability and myelination of peripheral axons by regulating SC function [17,18,19].

LKB1-mediated mitochondrial metabolism and lipogenesis are necessary for myelination during SC differentiation [17]. LKB1 deletion in SCs results in SC metabolic deregulation (e.g., mitochondrial dysfunction, energetic depletion, reduced lipid metabolism, and increased lactate) and subsequent axon degeneration [17]. Unlike other neuropathies often accompanied by inflammation and demyelination, LKB1-mediated axonal damage is characterized by disruption to cellular metabolisms. CNP-Cre-LKB1 cKO mice failed to activate mitochondrial oxidative metabolism during SC differentiation and reduced the production of citrate, a precursor to cellular lipids for myelination. These mutated mice exhibited PNS hypomyelination, neuropathy, and hindlimb dysfunction [17]. Restoring citrate partially rescued SC defects in LKB1-deleted mice. LKB1 deletion in SCs using P0 promoter further supports the critical role of LKB1 for maintaining axon integrity by mediating metabolic pathways [19]. P0-Cre-LKB1 cKO mice displayed the degeneration of myelinated and unmyelinated PNS axons starting at 3 months old, although PNS degeneration starts from a young age in CNP-Cre-LKB1 cKO mice. The main signaling pathways for LKB1-mediated metabolic alterations are probably independent of AMPK and mTOR signals [19] although further studies are required.

## 7. LKB1 Acts as a Molecular Target for CNS Axon Regeneration

A recent study demonstrates the crucial role of LKB1 in promoting the regenerative ability of mature CNS neurons and functional recovery after CNS axon injury in mammals, indicating LKB1 as a novel and promising target for neural repair [5]. Axon injury frequently results in persistent functional defects in various neurological disorders, but there are currently no treatments for functional deficiency due to CNS axon disconnection. It is very important to develop effective therapies for CNS damage. Many cell-autonomous molecules have been reported to manipulate the growth ability of mature neurons [82], but none of them have been translated to clinics yet. Thus, there is an unmet medical need to identify better targets for treating CNS lesions.

During development, LKB1 critically controls axon differentiation of some CNS neurons by regulating axon initiation and elongation [14,61], indicating that LKB1 may regulate regrowth and neuroplasticity after CNS axon injuries. LKB1 is downregulated in the cortices of developing mice, with the lowest level in adults [5], suggesting the potential correlation between reduced LKB1 levels and the growth ability of mature neurons in developed CNS. Several other genes regulating axon growth, including progranulin and phosphorylated mTOR, also exhibit age-dependent expression changes in the mouse brain [83,84]. LKB1 overexpression by AAV2 viral vector infection enhances neurite outgrowth of mature DRG neurons cultured on various inhibitory substrates, including the inhibitors derived from scar tissue and CNS myelin [5], suggesting that LKB1 signaling contributes to the growth failure of adult mammalian neurons (Figure 4).

Intracortical treatment with AAV2-LKB1 vector delivered several days after injury promotes the regrowth of descending corticospinal tract (CST) axons in adult mice [5]. AAV2-LKB1 vector was injected into the sensorimotor cortex 5 days after dorsal over-hemisection at T7 in adult C57BL/6 mice, and the regrowth of CST axons anterogradely traced by biotinylated dextran amine tracer was assessed 8 weeks after SCI. Upregulating LKB1 in the motor cortex stimulated the sprouting of CST axons rostral to the lesion, and remarkable CST regeneration into the lesion area and the caudal spinal cord. Many regenerated CST axons paralleled the GFAP-labeled reactive astrocytic processes surrounding the dorsal lesion epicenter and grew into the deeply transected areas close to the ventral spinal cord. CST axons in some mice regrew over 1 mm into the caudal spinal cord but reached >4 mm caudal to the lesion in other mice. Quantification of LKB1(+) cells co-localized with the neuronal marker NeuN in the motor cortex showed that >80% of neurons had strong LKB1 signals.

LKB1 upregulation in mature neurons by systemic treatments with a mutant AAV9 vector injected either before or after injury promoted robust regeneration of CST axons in adult mice [5]. The capsid protein of this AAV9 included two mutations by replacing two surface-exposed tyrosine residues with phenylalanine residues, which increased the BBB transference and infectivity of the viral vector [85]. Synapsin I promoter was used for selectively expressing LKB1 in neurons. Some mice treated with mutant AAV9-LKB1 exhibited long-distance CST regeneration into the lumbar enlargement, which contains motor neurons that supply the lower limbs. Double labeling for CSTs and a marker for glutamatergic axon terminals indicated the presence of synaptic terminals in regenerated CST axons at the lumbar spinal cord. Overexpressing of LKB1 by systemic mutant AAV9 vector also significantly increased the fiber length of other motor axons, including the descending serotonergic and tyrosine hydroxylase tracts at different spinal cord levels caudal to the lesion.

Somatic expression of LKB1 appears critical for controlling the regrowth of mature neurons because the somata of AAV2-LKB1-infected neurons exhibited strong LKB1 protein signals in vitro and in vivo [5]. By contrast, only a small portion of axons exhibited weak to moderate LKB1 signals in adult neurons. Regenerated CST axons in the caudal spinal cord of adult mice treated with AAV-LKB1 vectors did not show clear LKB1 signals.

Upregulating LKB1 by intracortical AAV2 or systemic mAAV9 vector treatment promotes significant functional recovery in SCI mice [5]. Evaluating motor function in mice with T7 dorsal over-hemisection SCI showed enhanced locomotor scores in either AAV2-LKB1- or mutant AAV9-LKB1-treated mice compared to SCI controls treated with the corresponding AAV for GFP. Most LKB1-treated mice also had better coordination than controls by more correctly placing their hind paws on grid walk bars 4 or 8 weeks after SCI. Systemic mutant AAV9-LKB1-treated mice also performed better in the touch-grasping test by increasing the hind limb grasping rate 4 and 8 weeks after SCI.

## 8. Potential Signal Pathways Downstream of LKB1 in Adult CNS

In the adult CNS, LKB1 appears to utilize multiple downstream signals, including AMPK1, NUAK1, and ERK signals, to convey its functions (Figure 5 and Table 1) [5]. The signaling pathways by which LKB1 controls the growth of mature CNS neurons were recently assessed in the motor cortices of adult mice two weeks after intracerebral AAV2-LKB1. Upregulating LKB1 in the mouse cortex increased the levels of p-AMPKα at Thr172 [5], thus probably activating AMPKα [86]. AMPK is known to regulate various neuronal functions, including synaptic remodeling in the retinas of aged mice [87], axogenesis, and axon growth during metabolic stress [88]. AAV2-LKB1 infection in the cortices of adult mice also significantly upregulated NUAK1, but not NUAK2. Consistently, the LKB1–NUAK1 pathway controls the branching of developing axons by immobilizing the axonal mitochondria [68]. In contrast, during neurodevelopment, LKB1 acts as a master upstream kinase for multiple signals controlling cell growth and axon formation and growth, mainly by activating NUAK1, SAD-A, and SAD-B kinases [15,50,68,77].

LKB1 upregulation in the adult cortex significantly activated Erk signals by increasing the levels of phosphorylated p44/42 MAPK (Erk1/2) at Thr202/Tyr204 [5]. ERK activation by viral infection also promoted CST regeneration following subcortical axotomy in adult rodents [89]. In contrast, AAV2-LKB1 reduced the expression levels of SAD-A, SAD-B, and phospho-S6 kinase [5] although all of them are essential for controlling axon differentiation and neuronal growth during development [14]. SADs represent a point of convergence whereby the PKA/LKB1 and PI3K/Akt pathways regulate axon formation by phosphorylating axon Tau on Ser-262 and organizing axon microtubule structures [15,50,77]. Suppressed S6 activity by AAV2-LKB1 in the adult cortex suggests that LKB1-stimulated regeneration of mature CNS neurons is largely independent of the mTOR/S6 kinase [5]. AAV2-LKB1 did not alter the levels of phosphorylated Akt and 4-E-BP1, two other critical signals along the PI3K pathways. In sum, LKB1 controls neural functions in adult CNS mainly by activating AMPKα, NUAK1, and ERK signals. In contrast, the PI3K signals (e.g., Akt, S6, and 4-E-BP1) and SAD-A and -B kinases play minimal roles following LKB1 activation in the adult CNS. Future studies may further dissect the detailed signaling pathways of LKB1 in mature neural cells.

## 9. Concluding Remarks

During development, LKB1 is a critical serine/threonine kinase controlling the metabolism, energy homeostasis, and polarity of various cell types by activating AMPK or AMPK-related kinases. In response to stimulation by extracellular factors, LKB1 acts as a major downstream effector of numerous signaling pathways and determines neuronal migration, axon initiation, and axon elongation during neurodevelopment. LKB1 is also involved in myelination by SCs during PNS development and subsequent maintenance of myelination. Importantly, a recent study supports the crucial role of LKB1 in mediating axon regeneration and neural repair after adult CNS injury.

Future studies are required to dissect additional functions of LKB1-dependent neural networks, such as synaptic integrity, neurodegeneration, neuronal survival, and the pathophysiology of various neurological disorders. LKB1 signaling maintains proper dendrite spacing in Purkinje cells and synaptic integrity in retinal neurons [87,90]. Reduced LKB1 activity may impair synaptic integrity and contribute to the pathogenesis of neurodegenerative disorders. Given that LKB1-dependent activity enhances glucose metabolism, mitochondrial biogenesis, and the survival of neural cells, reduced LKB1 levels may contribute to the loss of neural cells under certain conditions [91]. LKB1 signaling increases mitochondrial function and resistance to energy stress and protects neurons. Deleting LKB1 in certain mouse brain regions using the RIP2-Cre line, which targets β-cells in the pancreas and some neurons in the brain, showed axon degeneration in the spinal cord and paralysis of the hind limbs, suggesting a potential role of LKB1 in preventing the neurodegeneration of certain neuronal populations [92]. LKB1 function is probably variable depending on the cell types and conditions. LKB1 signaling promotes the differentiation and survival of developing hair cells [93], but facilitates hair cell death and synaptopathy due to the noise exposure [94].

Given the essential role of LKB1 in controlling SC polarity and PNS myelination, it would be interesting to dissect the potential roles of LKB1 signaling in regulating the functions of other types of glia. It is crucial to determine if LKB1 signaling controls the metabolism and function of oligodendrocytes and OPCs, and if LKB1 deficiency causes mitochondrial dysfunction and progressive axon degeneration of CNS neurons. LKB1 deficiency in astrocytes has been reported to reduce astrocytic metabolic function and enhance inflammatory responses [95]. The LKB1-AMPK signaling pathway may also modulate the actions of microglia under certain conditions [96].

Further research is required to dissect detailed signaling pathways of LKB1 in the adult nervous system, including its molecular and cellular mechanisms in regulating axon regeneration, and to compare the similarities and differences of LKB1 upstream and downstream signals between developmental and mature nervous systems. Given the unmet medical demands for functional regeneration and repair after CNS injury, it is extremely important to design feasible translational strategies for neuroplasticity and functional recovery by targeting LKB1 pathways, alone or in combination with other effective approaches.

## Figures and Tables

**Figure 1 cells-11-02861-f001:**
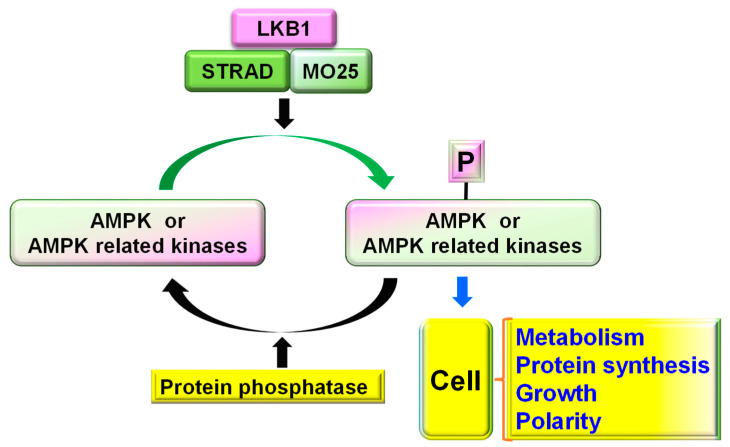
Schematic of LKB1–STRAD–MO25 complex and its major functions. LKB1–STRAD–MO25 complex phosphorylates AMPK and AMPK-related kinases, which can be dephosphorylated by protein phosphatases. Increased phosphorylation of AMPK and its related kinases controls various cellular functions, including cell energy metabolism, protein synthesis, growth, and polarity.

**Figure 2 cells-11-02861-f002:**
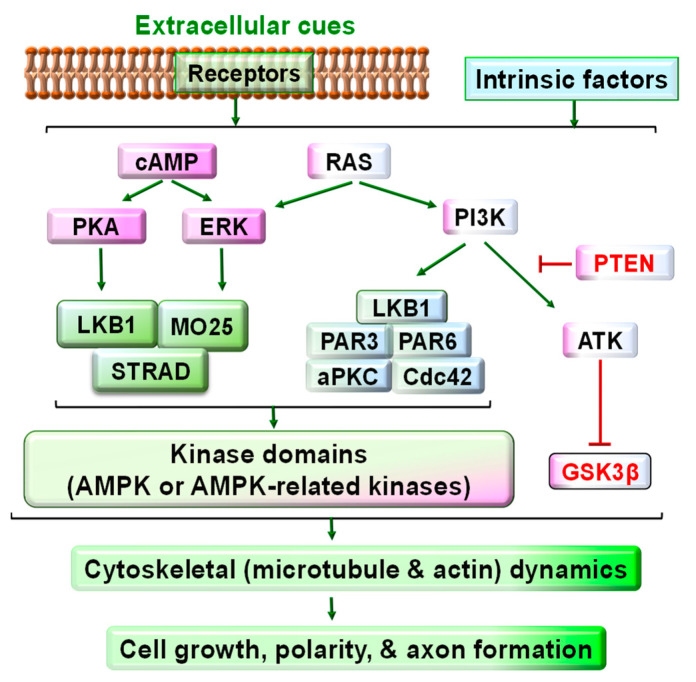
Schematic of the signaling pathways upstream and downstream of LKB1 in developmental neurons. Various extracellular cues and intracellular intrinsic factors activate many signaling pathways, including cAMP-PKA, cAMP/RAS-Erk, and RAS-PI3 kinase signaling. Downstream of these pathways, LKB1 is critical to convergent signaling for regulating cell growth. LKB1 usually forms a complex with STRAD and MO25 for LKB1 stability and activation. LKB1 may also form a complex with other proteins, including PAR3/PAR6/aPKC/Cdc42. As an intracellular master kinase, LKB1 is crucial for controlling cell growth, polarity, and axon formation and elongation during development by phosphorylating AMPK and AMPK-related kinases and regulating their activities accordingly. AMPK and AMPK-related kinases are associated with the cytoskeletal (such as microtubule and actin) dynamics and cell growth. Other LKB1-independent intracellular pathways, such as PI3K-Akt-mTOR signaling, may also modulate cell growth by diverse molecular mechanisms.

**Figure 3 cells-11-02861-f003:**
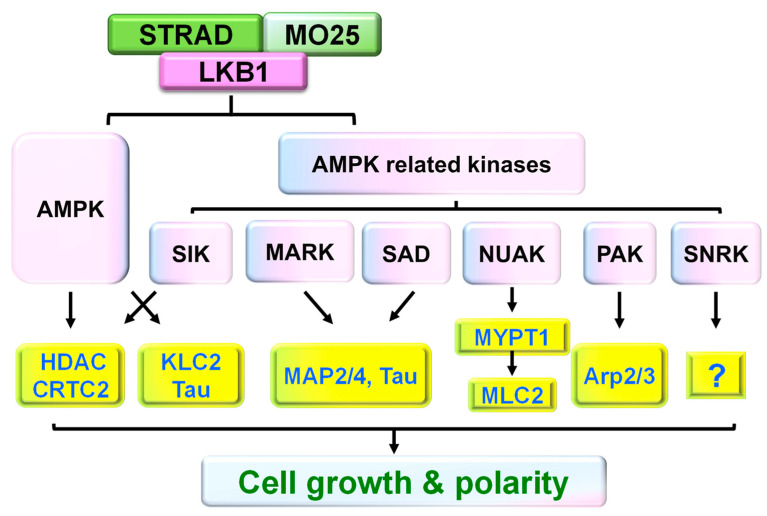
Schematic of phosphorylation of AMPK and AMPK-related kinases by LKB1 complex. LKB1 complex directly phosphorylates and activates AMPK and numerous AMPK-related kinases, including SAD kinases, NUAKs, MARKs, SIK1, and SNRK. These kinases then phosphorylate various downstream substrates to control several cellular functions, including cell growth and polarity. The downstream substrates include HADC (histone deacetylases), CRTC2 (CREB regulated transcription coactivator 2), KLC2 (kinesin light chain 2), Tau, MYPT2 (myosin phosphatase target subunit 2), and Arp2/3. MYPT2 may regulate cell growth through MLC2 (myosin light chain 2). So far, the substrate(s) of SNRK remains unknown.

**Figure 4 cells-11-02861-f004:**
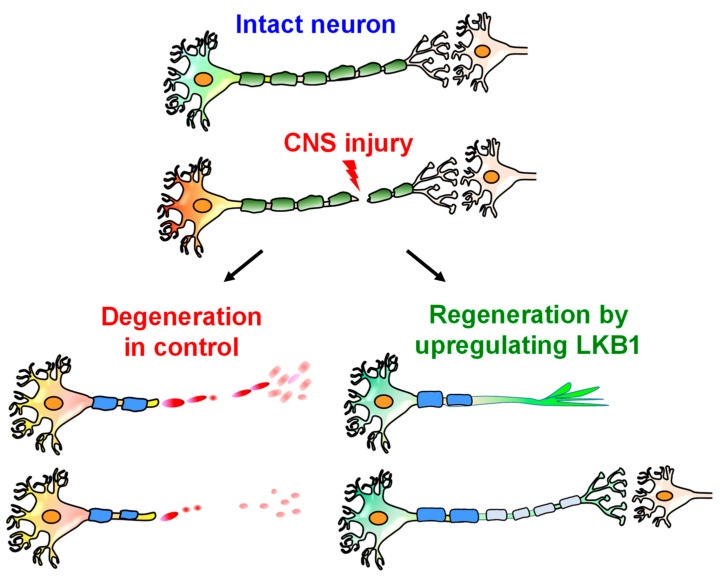
Schematic of axon regeneration of injured mature CNS neuron by upregulating LKB1. Axotomy in the CNS usually induces axon degeneration without obvious regrowth. In contrast, neuronal upregulation of LKB1promotes the regrowth of injured CNS axons in adult mammals.

**Figure 5 cells-11-02861-f005:**
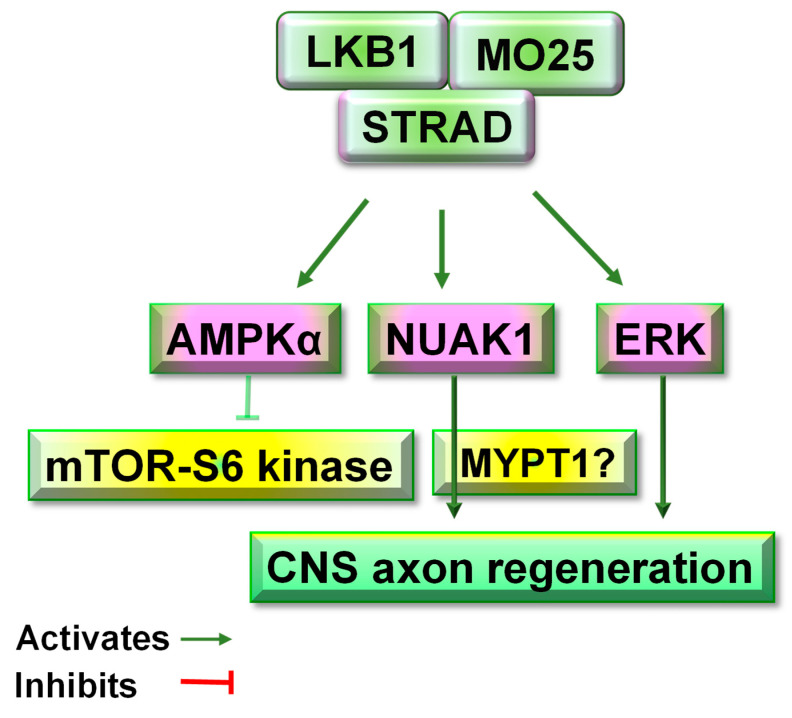
Schematic of the LKB1 signaling pathways in adult mammalian neurons. LKB1 upregulation in the cortical neurons of adult mice by AAV transduction increased the expression of NUAK1 and activated AMPKα and ERK signals. In contrast, LKB1 overexpression reduced the levels of SAD-A, SDA-B, and phosphorylated S6 kinase, and did not alter the levels of phosphorylated Akt and 4E-BP1 [5]. Thus, LKB1 probably regulates the function of mature neurons primarily by activating AMPKα, NUAK1, and ERK signals. It would be interesting to further dissect the comprehensive signaling of LKB1 in adult CNS, such as the possible role of MYPT1 (a myosin phosphatase) in regulating cytoskeletal dynamics downstream of NUAK1.

**Table 1 cells-11-02861-t001:** The effects of LKB1 on several downstream signaling proteins in developmental and mature neurons.

Proteins	Developmental Neurons	Mature CNS Neurons
AMPKα	Phosphorylate & increase activity [28,38]	Phosphorylate & increase activity [5]
NUAK1	Phosphorylate & increase activity [37]	Upregulate expression, likely increase its activity [5]
NUAK2	Phosphorylate & increase activity [37]	No change in expression [5]
SAD A	Phosphorylate & increase activity [15]	Reduce its expression [5]
SAD B	Phosphorylate & increase activity [15]	Reduce its expression [5]
Akt	Unknown	No change in phosphorylated Akt [5]
S6 kinase	Unknown	Decrease its phosphorylation & activity [5]
4E-BP1	Unknown	No change in phosphorylated 4E-BP1 [5]
ERK	Unknown	Increase its phosphorylation & activity [5]
SIK	Phosphorylate & increase activity [90]	Unknown
MARK	Phosphorylate & increase activity [47]	Unknown
PAK	Phosphorylate & decrease activity [56]	Unknown
SNRK	Phosphorylate & increase activity [60]	Unknown

## Data Availability

Not applicable.

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
