# Peer review of "Liver Kinase B1 Functions as a Regulator for Neural Development and a Therapeutic Target for Neural Repair"

_cells, 2022, doi:10.3390/cells11182861_

Round 1

Reviewer 1 Report

This review comprehensively summarized the major cellular and molecular mechanisms of the Liver kninase B1 (LKB1) in neuronal polarization and axonal initiation during the development of CNS. It also discussed the roles of LKB1 in promoting neural repair and axonal regeneration and enhancing its expression as a as a potential therapeutic approach for neurological diseases, such as spinal cord injury. Furthermore, it provided very useful insights for future studies to dissect the molecular mechanisms of LKB1 in neural repair after injury in adult CNS. This well written and very informative review could be of broad interest to many readers of Cells, such as neuroscientists, molecular biologists, et al.

 There are a few minor concerns: 1) highlighting the subtitles, e.g. numbering and bold font, will make the review much easier to follow; 2) the downstream pathways of LKB1 in neural cells during development are comprehensively. Adding a figure/scheme to illustrate these pathways and their interactions will increase the readability; 3) the axonal regeneration shown in Figure 2 is a little bit confusing and misleading. The illustration in figure 2 could be misunderstood that regeneration is to reconnect the uninjured side with the injured distant side. However, after injury, the axon distant to injury degenerate/demyelination/die. It will be clearer to show: a) the uninjured normal axons, b) the degenerating axons after injury and c) the regenerating axons, using different colors/shadows; 4) on page 6, line 183, the underline followed ref#60 should be deleted.  

Author Response

We greatly appreciate the excellent comments and suggestions.  We have revised the paper as advised here.

1) Format issues: The format issues raised were caused during the submission to Cells. We have corrected these in this resubmission.

2) Adding a figure/scheme to illustrate the pathways during development:  We have added a new figure (Figure 3) to illustrate the kinases targeted by LKB1 during development.

3) Confusing illustration in figure 2:  We have revised the illustration in figure 2, as suggested by this reviewer. It becomes Figure 5 in the current version.

Reviewer 2 Report

Line 91-92 “ functions. Among the diverse signals downstream of these pathways, phosphorylating LKB1 at different sites is crucial for cell functions”, please provide more details of that different cell functions respond to of different site of LKB1 phosphorylated.  

In line 114-115, “Human STRAD includes two isoforms, STRAD α and β. Both isoforms have a STE20-like kinase domain but lack the residues critical for catalytic activity (termed pseudokinase).” This sentence is not necessary, consider this paragraph is talking about STRAD affect kinase activity of LKB1

In line 130 “PKC. also mediates the suppression of scar-sourced axon growth inhibitors by LAR receptors”, I am confused about the relationship among PKC, LAR and LKB1. Please clarify.

In line 131 “Other signaling proteins, such as PTEN [36], may also interact with LKB1 and modulate its localization and activity”. Please provide more details.

Between line 133 and line 180. The subtopic is “Major signaling pathways downstream of LKB1 in neural cells”. However, many literatures cited here are not from neurons but cancer cells. Please double check.

It will be best to see a table compare the difference of LKB1 between young and mature neurons.

Author Response

We thank this reviewer for the excellent comments and suggestions.  We have revised the paper as advised here.

1) Please provide more details of that different cell functions respond to of different site of LKB1 phosphorylated.  

We have now provided the information on LKB1 phosphorylation sites and their functions in the second paragraph of Section 2 (Gene, structure, and overall cellular functions).

2) “Human STRAD includes two isoforms, STRAD α and β. Both isoforms have a STE20-like kinase domain but lack the residues critical for catalytic activity (termed pseudokinase).” This sentence is not necessary, consider this paragraph is talking about STRAD affect kinase activity of LKB1.

We have removed this sentence.

3) PKC-zeta also mediates the suppression of scar-sourced axon growth inhibitors by LAR receptors”, I am confused about the relationship among PKC, LAR and LKB1.

We have rewritten this sentence to clarify.  “Activating LAR receptors by CSPG application reduces the activities of PKCζ and LKB1, indicating that PKCζ-mediated LKB1 suppression also mediates axon growth inhibition by the scar-scoured inhibitor CSPGs [35].”

4) “Other signaling proteins, such as PTEN [36], may also interact with LKB1 and modulate its localization and activity”. Please provide more details.

We have rewritten this sentence. “Other signaling proteins may also interact with LKB1 and modulate its relocation and activity. For example, the interactions between LKB1 and PTEN have been shown to promote LKB1 relocation to the cytoplasm in cancer cells [36].”.

5) The subtopic is “Major signaling pathways downstream of LKB1 in neural cells”. However, many literatures cited here are not from neurons but cancer cells. Please double check.

Most signaling pathways downstream of LKB1 were studied in neural cells, but some of them were performed in cancer cells.  We thus have changed the subtitle to “4. Major signaling pathways downstream of LKB1 in neurons or other cell types”.

6) It will be best to see a table compare the difference of LKB1 between young and mature neurons.

We have added Table 1 to compare the effects of LKB1 on some signaling proteins.

Reviewer 3 Report

This review provides useful information about the function of liver kinase B1 in neural growth and repair. LKB1 is crucial for regulating neuronal polarization and axon branching throughout development, hence the subject is regarded as being of high interest. However, there are still a lot of points that need improvement.

1-     If the existing text is determined to be readable and coherent, I advise including a title or even a subtitle for each section.

2-     The review's general structure needs significant improvement. The structure of LKB1 should generally come first, followed by an explanation of its function in physiology or development, and finally, its significance in neuronal injury. These points might strengthen and improve the review.

3-     The LKB1 structural picture must be the first figure in the text.

4-     It is important to emphasize LKB1's function in preventing neurodegeneration.

Author Response

We greatly appreciate the great comments and suggestions.   

1) Including a title or even a subtitle for each section.: Loss of subtitles occurred during the submission to Cells. We have corrected this issue in this resubmission.

 2) General structural improvement.

 We have reorganized the figures extensively and moved the LKB1 structure to Figure 1.  We feel that the current contents in all Sections flow well. 

3) It is important to emphasize LKB1's function in preventing neurodegeneration.

We have added information on this topic in the last Section of this review (9. Concluding remarks). “Deleting LKB1 in certain mouse brain regions using RIP2-Cre line, which targets β-cells in the pancreas and some neurons in the brain, showed axon degeneration in the spinal cord and paralysis of the hind-limbs, suggesting a potential role of LKB1 in preventing neurodegeneration of certain neuronal populations [92].”

Notably, we did not include the details in previous version because the study of 92 used the RIP2 promoter, which mainly targets β-cells in the pancreas.

Reviewer 4 Report

The paper is well prepared about Liver Kinase B and related interacting protein networks in the neuronal cells. However, the given evidences for the neuronal cells for the classification of the LKB are missing; current data released on databases is not included and not mentioned at the required level. Therefore there is no significant remark on the targeting LKB for this perspective. This is review work, authors should also give more detail about how the funding agency supported the literature review. More experimental evidence should be on the subject by increasing figures numbers to highlight molecular phenomena. 

Author Response

We thank this reviewer for the comments and suggestions.

1) Current data released on databases is not included.  We are sorry for unable to addressing the issue because we do not know which databases this reviewer referred to. We believe that our subtitles and figures cover the reviewed topics well, as indicated by the other 3 reviewers.

2) To clarify funding support: We have further clarified the funding support information.

3) Extend the figures: We have extended the previous two figures. The current version has five figures.

Round 2

Reviewer 3 Report

The authors address each of my earlier concerns in this review article that has been updated. After some minor language editing, the article is now suitable for publication.